

# HIV-1 promoter is gradually silenced when integrated into *BACH2* in Jurkat T-cells

Anne Inderbitzin[1,2,3,*], Yik Lim Kok[1,2,*], Lisa Jörimann[1,2,3],
Audrey Kelley[1,2,3], Kathrin Neumann[1,2], Daniel Heinzer[4,5],
Toni Cathomen[6,7] and Karin J. Metzner[1,2]

[1] Department of Infectious Diseases and Hospital Epidemiology, Division of Infectious Diseases and Hospital Epidemiology, University Hospital Zurich, Zurich, Switzerland
[2] Institute of Medical Virology, University of Zurich, Zurich, Switzerland
[3] Life Science Zurich Graduate School, University of Zurich, Zurich, Switzerland
[4] Institute for Neuropathology, University Hospital Zurich, Zurich, Switzerland
[5] Neuroscience Center Zurich Graduate School, University of Zurich, Zurich, Switzerland
[6] Institute for Transfusion Medicine and Gene Therapy, Medical Center, University of Freiburg, Freiburg, Germany
[7] Faculty of Medicine, University of Freiburg, Freiburg, Germany
* These authors contributed equally to this work.

## ABSTRACT

**Background:** The persistence of the latent HIV-1 reservoir is a major obstacle to curing HIV-1 infection. HIV-1 integrates into the cellular genome and some targeted genomic loci are frequently detected in clonally expanded latently HIV-1 infected cells, for instance, the gene *BTB domain and CNC homology 2 (BACH2)*.
**Methods:** We investigated HIV-1 promoter activity after integration into specific sites in *BACH2* in Jurkat T-cells. The HIV-1-based vector LTatCL[M] contains two fluorophores: (1) Cerulean, which reports the activity of the HIV-1 promoter and (2) mCherry driven by a constitutive promotor and flanked by genetic insulators. This vector was inserted into introns 2 and 5 of *BACH2* of Jurkat T-cells via CRISPR/Cas9 technology in the same and convergent transcriptional orientation of *BACH2*, and into the genomic safe harbour AAVS1. Single cell clones representing active (Cerulean+/mCherry+) and inactive (Cerulean−/mCherry+) HIV-1 promoters were characterised.
**Results:** Upon targeted integration of the 5.3 kb vector LTatCL[M] into *BACH2*, the HIV-1 promoter was gradually silenced as reflected by the decrease in Cerulean expression over a period of 162 days. Silenced HIV-1 promoters could be reactivated by TNF-α and Romidepsin. This observation was independent of the targeted intron and the transcriptional orientation. *BACH2* mRNA and protein expression was not impaired by mono-allelic integration of LTatCL[M].
**Conclusion:** Successful targeted integration of the HIV-1-based vector LTatCL[M] allows longitudinal analyses of HIV-1 promoter activity.

Corresponding author
Karin J. Metzner,
Karin.Metzner@usz.ch

## INTRODUCTION

Antiretroviral therapy (ART) efficiently blocks virus replication; however, it does not cure HIV-1 infection due to the presence of replication-competent but silenced proviruses preferentially integrated in long-lived resting CD4$^+$ T-cells (*Finzi et al., 1997*; *Wong et al., 1997*). Various factors and molecular mechanisms that result in HIV-1 latency have been proposed (*Ruelas & Greene, 2013*). One such factor might be the integration site of the provirus, which has been suggested to not only be responsible for silencing the provirus, but also supporting cell expansion, thus maintaining the size of the HIV-1 latent reservoir (*Maldarelli et al., 2014*; *Wagner et al., 2014*). Integration of HIV-1 into the human genome is not random. In vivo and ex vivo HIV-1 integration site analyses revealed that HIV-1 favours integration into introns of active transcription units in gene-dense regions, although a minority of integration events outside of these regions have consistently been observed (*Ciuffi & Bushman, 2006*; *Kok et al., 2016*; *Mitchell et al., 2004*; *Schroder et al., 2002*; *Stevens & Griffith, 1996*). Furthermore, HIV-1 appears to target active transcription units that are in close proximity to the nuclear pore (*Marini et al., 2015*). On the population level, intragenic HIV-1 does not have a preference for either transcriptional orientation of the targeted gene (*Cohn et al., 2015*; *Schroder et al., 2002*; *Stevens & Griffith, 1996*). HIV-1-infected cells can undergo clonal expansion (*Cesana et al., 2017*; *Cohn et al., 2015*; *Maldarelli et al., 2014*; *Rezaei & Cameron, 2015*; *Satou et al., 2017*; *Wagner et al., 2014*), and the proviruses in these clonally expanded cells are often located in specific regions of the human genome. One recurrent integration gene (RIG) that has been observed across patients in numerous independent studies is the gene *BACH2*, in which the provirus is almost exclusively found in intron 5 and in the same transcriptional orientation as *BACH2* (*Cesana et al., 2017*; *Ikeda et al., 2007*; *Imamichi et al., 2014*; *Mack et al., 2003*; *Maldarelli et al., 2014*; *Wagner et al., 2014*). Since these *BACH2* integration sites were identified in HIV-1-infected individuals who have been on ART for several years, it is conceivable that these proviruses are inactive, although it remains unknown whether this presumed inactivity is a result of integration site-dependent silencing of replication-competent proviruses or due to defective proviruses.

To address the question of whether the HIV-1 promoter would be silenced upon integration into intron 5 of *BACH2* in the same transcriptional orientation, we employed a modified version of our dual-fluorophore HIV-1-based vector, LTatC[M], which reproduces features of active and latent HIV-1 infections (*Kok et al., 2018*). This vector comprises two fluorescent reporter genes: (1) Cerulean, which reports the activity of the HIV-1 promoter and (2) mCherry, the expression of which is driven by a constitutive promoter and further protected from position-effect variegation by a pair of flanking genetic insulators to identify cells harbouring an integrated vector (*Uchida et al., 2013*; *Villemure, Savard & Belmaaza, 2001*; *Yahata et al., 2007*). In this study, we investigate whether CRISPR/Cas9-mediated targeted HIV-1 integration in *BACH2* is feasible and would lead to inactivation of the HIV-1 promoter over time, and if so, whether it is locus and/or transcriptional orientation dependent.

## MATERIALS AND METHODS

### Generation of LTatCL[M] with target locus homologous arms and Cas9/guide RNA-encoding plasmids

In the HIV-1 based, dual-fluorophore vector LTatC[M] the 3′LTR is located downstream of the second fluorophore mCherry to enable retrovirus production and subsequent infection of target cells (Kok et al., 2018). LTatC[M] was modified to LTatCL[M], that is, the 3′LTR (L) was inserted between Cerulean (C) and the insulator cHS4 (Fig. 1A) to further enhance the transcriptional independence of the HIV-1 promoter controlled Cerulean. For targeted integration of this HIV-1 based, dual-fluorophore vector, retrovirus production is not required. Thus, the HIV-1 3′LTR was relocated immediately downstream of Cerulean. Additionally, a polyA signal was inserted between mCherry ([M]) and the second insulator sMAR8 (Fig. 1A). The homologous regions on both sides of the targeted HIV-1 integration site in the human genome were obtained from NCBI GenBank: BACH2 intron 5 (Accession No: NT_007299.13; 5′ arm nucleotides 93′502–94′355, 3′ arm nucleotides 94′356–95′206), BACH2 intron 2 (5′ arm nucleotides 339′363–340′186, 3′ arm nucleotides 340′187–341′034) and AAVS1 (Accession No: NC_000019.10; 5′ arm nucleotides 1′399–2′218, 3′ arm nucleotides 2′219–3′051). Targeted integration sites are depicted in Fig. 1B.

Primers to amplify the respective target locus homologous arms are listed in Table S1. Each PCR reaction contained 100 ng of human genomic DNA (Sup-T1 cell line; obtained through the NIH AIDS Reagent Programme, Division of AIDS, NIAID, NIH, from Dr. Dharam Ablashi), 1x Platinum Taq PCR buffer (ThermoFisher, Waltham, MA, USA), 2 mM $MgCl_2$ (ThermoFisher, Waltham, MA, USA), 0.2 mM dNTP (NEB), 0.4 μM of each forward and reverse primer and 1 U Platinum Taq polymerase (ThermoFisher, Waltham, MA, USA) in a total volume of 50 μL. The PCR cycling conditions were as follows: 94 °C for 2 min; 35 cycles of (94 °C for 30 s, 55 °C for 30 s, 68 °C for 1 min); 68 °C for 5 min; 4 °C hold.

The respective pair of target locus homologous arms were cloned into pGEM-T Easy (Promega). Subsequently, the dual-fluorophore vector LTatCL[M] was cloned into each plasmid in the same or convergent orientation via blunt-end cloning. The 6 plasmids pBACH2_i5-, pBACH2_i2- and pAAVS1-LTatCL[M] were generated, containing the homologous arms of the targeted HIV-1 integration loci in BACH2 intron 5, BACH2 intron 2, and in the genomic safe harbour AAVS1, respectively, containing LTatCL[M] in both transcriptional orientations (s and c) (Fig. 1A).

Plasmids encoding Cas9 and guide RNAs were based on PX458 (Table S1). pSpCas9 (BB)-2A-GFP (PX458) was a gift from Feng Zhang (Addgene plasmid # 48138; http://n2t.net/addgene:48138; RRID:Addgene_48138) (Ran et al., 2013). Annealing of 100 μM 5′ phosphorylated primers for the guide RNAs was performed using the following conditions: 80 °C for 5 min, 65 °C for 7 min, 60 °C for 7 min, 55 °C for 7 min, 50 °C for 7 min, 45 °C for 7 min, 40 °C for 7 min, 35 °C for 7 min, 30 °C for 7 min, 25 °C for 7 min and 4 °C hold. The annealed guide RNA primers were separately cloned into PX458.

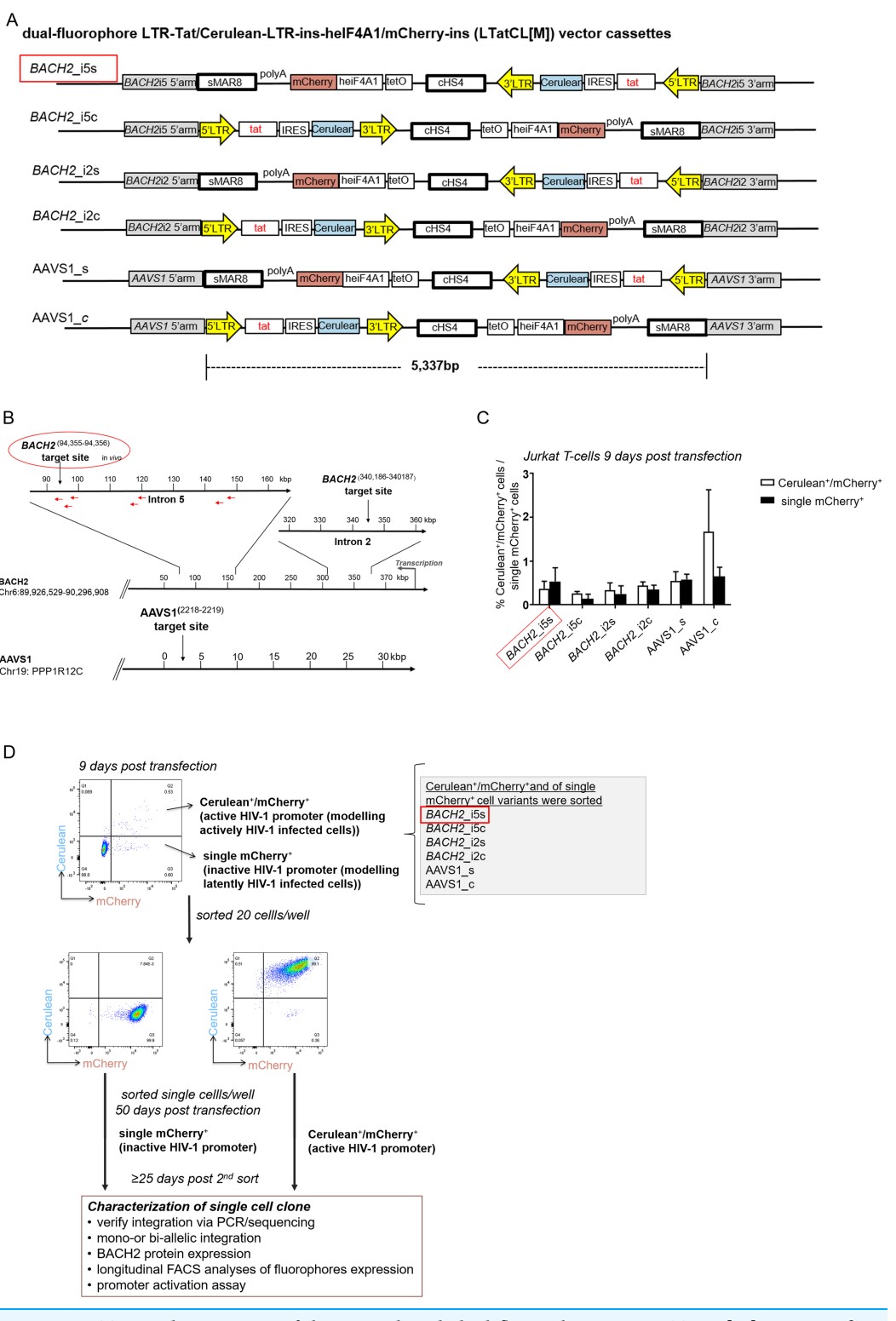

**Figure 1 Targeted integration of the HIV-1 based, dual-fluorophore vector LTatCL[M] into specific genomic loci in Jurkat T-cells.** (A) Schematic diagram of the six HIV-1 based, dual-fluorophore vectors LTatCL[M] (5'337 bp) flanked with the *BACH2*/AAVS1 homologous arms. LTatCL[M] contains two fluorophores (Cerulean and mCherry) to distinguish between inactive and active HIV-1 promoters, that is modelling latently and actively HIV-1 infected cells. The Cerulean cassette is under the control of the

**Figure 1** (continued)
HIV-1 LTR whereas mCherry is under the control of an independent constitutive promoter (heIF4A1) and flanked by two insulators (cHS4 and sMAR8). LTR, long terminal repeat; tat, HIV-1 transactivator; IRES, internal ribosome entry site; cHS4, chicken hypersensitive site 4; tetO, tetracycline operator sequences; heIF4A1, human eukaryotic initiation factor 4A1, gene promoter; sMAR8, synthetic matrix attachment region 8. (B) Scheme of the targeted HIV-1 integration sites in *BACH2* and AAVS1. Some described HIV-1 integration sites in vivo are marked by red arrows (*Maldarelli et al., 2014*; *Wagner et al., 2014*). (C) Percentage of Cerulean[+]/mCherry[+] (white bars) and single mCherry[+] (black bars) cells 9 days post transduction of Jurkat T-cells targeting the different loci in *BACH2* and AAVS1. The means and standard deviations of 3 independent experiments are depicted. (D) Flow chart to generate monoclonal cell lines. Jurkat T-cells were separately transfected with the vectors shown in A and the corresponding gRNA/Cas9 plasmid. Nine days post transfection, the six different targeted HIV-1 integration variants were each sorted by 20 cells per well for the phenotypes Cerulean[+]/mCherry[+] and single mCherry[+]. Fifty days post transfection, the cells were single cell sorted for each targeted integration variant for the two phenotypes Cerulean[+]/mCherry[+] and single mCherry[+]. After at least 25 days post second sorting, cells were further characterised. (A–D). The in vivo observed preferential HIV-1 integration loci in *BACH2*, *BACH2*_i5s, is highlighted by red boxes.

All plasmid sequences were confirmed by next-generation sequencing. Briefly, plasmids were gel purified, diluted to 0.2 ng/µl, and 1 ng of the plasmids were processed using the Illumina Nextera XT DNA library Prep kit. The DNA libraries were subsequently paired-end or single-end sequenced with an Illumina MiSeq using the MiSeq Reagent Kit v3 (150-cycle). Constructs are shown in Fig. 1A.

## Transfection of Jurkat T-cells and single cell sorting

Prior to transfection, the six LTatCL[M] vectors (p*BACH2*_i5-, p*BACH2*_i5c-, p*BACH2*_i2s-, p*BACH2*_i2c-, pAAVS1_s-, and pAAVS1_c- LTatCL[M]) were linearised using 100 units of NsiI (NEB)/20 µg DNA. For transfection, 1 million Jurkat T-cells were resuspended in 100 µL Nucleofector solution (Cell Line Nucleofector[TM] Kits, Lonza) and combined with 2 µg of the linearised dual-fluorophore vector and 2 µg of the corresponding gRNA/Cas9 plasmid: PX458_g*BACH2*[94,355–94,356] (intron 5), PX458_g*BACH2*[340,186–340,187] (intron 2) and PX458_gAAVS1[2,218–2,219] (AAVS1). Nucleofection was performed using the programme X-001 for Jurkat T-cells (Amaxa[TM] Nucleofector[TM] II, Lonza). Cells were cultured in RPMI-1640 medium media supplemented with 10% fetal bovine serum (FBS) and 1% penicillin/streptomycin (10,000 units/ml penicillin, 10 mg/ml streptomycin).

At day 9 post transfection, cells were analysed using fluorescence-activated cell sorting (FACS) and sorted at 20 cells per well in 96-well plates using a BD FACSAria[TM] III (BD Biosciences, Franklin Lakes, NJ, USA). Two cell phenotypes were sorted: 1. Cerulean[+]/mCherry[+] and 2. Cerulean[−]/mCherry[+]. This was done for each nucleofected cell population: *BACH2*_i5s, *BACH2*_i5c, *BACH2*_i2s, *BACH2*_i2c, AAVS1_s, and AAVS1_c. After 50 days, expanded cell cultures were analysed by flow cytometry and subsequently single cell sorted. Longitudinal flow cytometric analysis of the cells were done with the LSRFortessa II (BD Biosciences, Franklin Lakes, NJ, USA) and data were analysed using the FlowJo Software v.10.0.8. (FLOWJO, LLC) (Fig. 1C).

## Activation of silenced HIV-1 promoter in cells with integrated LTatCL[M] in BACH2

Over time silenced HIV-1 promoter in sorted Cerulean[+]/mCherry[+] monoclonal cell lines were reactivated using 10 ng/μL tumor necrosis factor alpha (TNF-α) and 4 nM Romidepsin (Ro) (Selleckchem, Houston, TX, USA). After 24 hrs the cells were analysed by flow cytometry.

## Amplifying the junctions of LTatCL[M] integration into BACH2 and AAVS1

Genomic DNA was extracted from 5 million transfected Jurkat T-cells using the DNeasy Blood and Tissue kit (Qiagen, Hilden, Germany) and quantified by Quant-iT[TM] PicoGreen[TM] dsDNA Assay kit (ThermoFisher Scientific, Waltham, MA, USA). To verify the targeted integration of the dual-fluorophore vector LTatCL[M] into *BACH2* or AAVS1, the junctions of targeted integration were amplified by (semi-)nested PCR using 300 ng genomic DNA containing 1x PCR buffer (Sigma–Aldrich, St. Louis, MI, USA), 1.5 mM $MgCl_2$, 0.2 mM dNTPs, 0.5 μM of each respective forward and reverse primer (Table S1), and 0.5 U JumpStart Taq DNA polymerase (Sigma–Aldrich, St. Louis, MI, USA) in a total volume of 25 μL. The PCR cycling conditions were as follows: 94 °C for 2 min; 35–40 cycles of (94 °C for 30 s, 63 °C for 30 s, 72 °C for 2 min); 72 °C for 5 min; 4 °C hold.

Mono-allelic or bi-allelic integration of LTatCL[M] into *BACH2* and AAVS1 was verified by PCR containing 100 ng genomic DNA, 1x PCR buffer, 1.5 mM $MgCl_2$, 0.2 mM dNTPs, 0.5 μM of each respective forward and reverse primers (Table S1), and 0.5 U JumpStart Taq DNA polymerase in a total volume of 25 μL. The PCR cycling conditions were as follows: 94 °C for 2 min; 35 cycles (94 °C for 30 s, 63 °C for 30 s, 72 °C for 2 min); 72 °C for 5 min; 4 °C hold.

Amplicons were sequenced by Sanger sequencing. The sequencing reaction contains 1x Seqmix (Big Dy Termination v1.1, 1x Dilution Buffer (ThermoFisher, Waltham, MA, USA)), and 100 ng DNA in a total volume of 40 μL. 0.2 μM primer was added (Table S1). The PCR cycling condition were as follows: 40 cycles (96 °C for 30 s, 50 °C for 30 s, 60 °C for 4 min); 4 °C hold. Sequencing was performed following the manufacturer's instruction on an 3130xl Genetic Analyzer (Applied Biosystems, Foster City, CA, USA).

## Near full-length amplification of integrated vector LTatCL[M]

Genomic DNA was extracted from 3 million transfected cells using the DNeasy Blood and Tissue kit and quantified by Quant-iT[TM] PicoGreen[TM] dsDNA Assay kit. To investigate the integrity of the integrated vector LTatCL[M], near full-length amplification of the vector was performed, using genomic 300 ng DNA, 1x Long Amp *Taq* Reaction Buffer (NEB), 0.4 mM dNTPs, 0.5 μM of each forward and reverse primer (Table S1) and 2.5 U Long Amp Taq Polymerase (NEB) in a total volume of 25 μL. The PCR cycling conditions were as follows: 94 °C for 30 min; 30 cycles of (94 °C for 20 s, 58–61 °C for 30 s, 72 °C for 5 min); 72 °C for 10 min; 4 °C hold. A total of 1 ng purified amplicons were

processed with the Nextera XT DNA Library Preparation Kit (Illumina) and subsequently sequenced using the MiSeq reagent Kit v2 as described above.

## Quantification of BACH2 mRNA expression

RNA was extracted from 3 million transfected cells using the All Prep DNA/RNA kit (Qiagen, Hilden, Germany) and quantified by Nanodrop 1000. A total of 800 ng RNA was reverse transcribed with Prime Script Reverse Transcriptase (Takara, Kyoto, Japan) according to manufacturer's instructions. To exclude contamination of plasmid and genomic DNA contamination, reactions lacking reverse transcriptase were included for each RNA sample. To quantify the mRNA level of $BACH2$, qPCR reactions were performed in triplicates for each sample. cDNA was diluted 1:10 mixed with 1x PCR buffer (Sigma–Aldrich, St. Louis, MI, USA), 1.5 mM $MgCl_2$, 0.2 μM dNTP (NEB), 0.5 μM of each Fw and Rv primers, 1x SYBR® Green, 50 nM Rox, 0.5 U JumpStart Taq Polymerase (Sigma–Aldrich, St. Louis, MI, USA) in a total volume of 25 μL. For each reaction, primer pair was chosen to amplify a specific region of the $BACH2$ mRNA, spanning exon 7–8. As a mRNA expression control glycerinaldehyd-3-phosphat-dehydrogenase (GAPDH) was amplified with GAPDH-specific primers (Table S1). The qPCR cycling conditions were as follows: 94 °C for 2 min; 40 cycles (94 °C for 30 s, 55 °C for 30 s, 63 °C for 1 min (collect data)); 72 °C for 2 min; 95 °C for 15 s; 4 °C hold. Melt curves were collected to analyse specificity of the amplification. The qPCR was performed in the ABI 7800 real-time PCR thermos-cycler then analysed using the 7500 Software v2.0.4, using the comparative $C_t$ method (*Livak & Schmittgen, 2001*).

## Quantification of BACH2 protein expression

Cells were lysed using assay lysis buffer (50 mM Tris-HCL pH8, 150 mM NaCl, 0.5% Na deoxycholate, 0.5% Triton X-100, 1x Protease inhibitor, $ddH_2O$) and protein was quantified using the Bicinchoninic acid assay (BCA) (Pierce™ BCA Protein Assay Kit). A total of 40 μg protein were mixed with loading dye and loaded on a Bolt™ 4–12% Bis–Tris Plus Gels (ThermoFisher Scientific, Waltham, MA, USA). Subsequently, proteins were transferred onto a nitrocellulose membrane (iBlot™ 2 Transfer Stacks, ThermoFisher Scientific, Waltham, MA, USA) blocked for 1 h at RT with 5% Top-Block wt/vol in PBS supplemented with Tween-20 (10x PBS pH7.4, 0.1% Tween 20, and $ddH_2O$). For protein detection, the membrane was incubated with primary antibody (1/400 diluted rabbit anti-BACH2 antibody (PA5-23642; ThermoFisher Scientific, Waltham, MA, USA) and 1/5,000 diluted rabbit-anti-GAPDH antibody (ab9485, Abcam, Cambridge, UK)) overnight at 4 °C. After washing the membrane four times with 1x PBS-T, the secondary antibody, 1/10'000 diluted goat anti-rabbit IgG H&L (HRP) (ab97051; Abcam, Cambridge, UK) was added and the membrane was incubated at RT for 1 h. The membrane was then washed four times with 1x PB-T. To visualise the protein, Immobilon Crescendo Western HRP substrate (Merck, Kenilworth, NJ, USA) was added to the membrane. Chemiluminescence was captured with the Stella system (model 3200; Matlab Group Companies, Natick, MA, USA).

## RESULTS

### Targeted integration of the 5.3 kb HIV-1-based, dual-fluorophore vector LTatCL[M] into BACH2 and AAVS1 via CRISPR/Cas9-mediated homology directed repair

To investigate specific effects of HIV-1 integration into *BACH2*, four constructs were generated containing the vector LTatCL[M] flanked by *BACH2* homologous arms (Fig. 1A) and integrated in Jurkat T-cells by means of the CRISPR/Cas9 technology. The constructs were integrated into intron 5 of *BACH2* in (1) the in vivo observed same (s) and (2) the convergent (c) transcriptional orientation to investigate the potential impact of transcriptional orientation on HIV-1 promoter activity. Furthermore, LTatCL[M] was integrated into intron 2 of *BACH2* in (3) the same and (4) the convergent transcriptional orientation to investigate the position effect of HIV-1 integration into *BACH2*. Intron 2 was chosen as it has not yet been observed in HIV-1 infected patients, however, it was frequently observed as HIV-1 integration locus in human CD34$^+$ haematopoietic stem cells infected in vitro with HIV-1(*Maldarelli et al., 2014*). In addition, LTatCL[M] was integrated into intron 1 of *PPP1R12C* (protein phosphatase 1 regulatory subunit 12C) in again both transcriptional orientations. This genomic locus was previously identified as preferred site for integration of adeno-associated virus (AAV) DNA and designated AAVS1 (AAV integration site) (*Kotin, Linden & Berns, 1992*). AAVS1 is widely used as genomic safe harbour (GSH) although not without some limitations particularly in the context of future use in gene therapy trials (*Papapetrou & Schambach, 2016*; *Sadelain, Papapetrou & Bushman, 2011*). Nevertheless, for research purposes it is considered a GSH and we expect that the HIV-1 promoter will not be silenced when integrated in this GSH. These six vector constructs and subsequently generated Jurkat T-cell clones were named *BACH2*_i5s, *BACH2*_i5c, *BACH2*_i2s, *BACH2*_i2c, AAVS1_c, and AAVS1_s (Figs. 1A and 1B).

Jurkat T-cells were nucleofected with linearised plasmids of the six LTatCL[M] constructs. Nine days post transfection, the frequencies of stably transfected Cerulean$^+$/mCherry$^+$ and single mCherry$^+$ cells were between 0.1% and 1.8% (Fig. 1C). Cells were sorted to enrich Cerulean$^+$/mCherry$^+$ (i.e., active HIV-1 promoter) and single mCherry$^+$ (i.e., inactive HIV-1 promoter) cells (Fig. 1D).

To obtain monoclonal cell lines, a sequential flow cytometric sorting strategy was employed. For all targeted integration variants, Cerulean$^+$/mCherry$^+$ and single mCherry$^+$ cells were first sorted at 20 cells per well in 96-well plates and expanded in culture up to 50 days post transfection. We analysed by means of flow cytometry a total of 92 cell populations, and selected 29 for further expansion. These cells were single-cell sorted to obtain monoclonal cell lines and expanded in culture for at least 25 days prior to further characterisation. In total, 43 single cells expanded resulting in 43 monoclonal cell lines. For each targeted integration variant (*BACH2*_i5s, *BACH2*_i5c, *BACH2*_i2s, *BACH2*_i2c, AAVS1_s and AAVS1_c), at least one monoclonal cell line for each of the two phenotypes (Cerulean$^+$/mCherry$^+$ and single mCherry$^+$) were obtained with the exception of AAVS1_c, single mCherry$^+$ (Table 1). The monoclonal cell lines were characterised as follows.

**Table 1 Characteristics of monoclonal Jurkat T-cell lines after targeted integration of LTatCL[M] into BACH2 and AAVS1.** Cerulean⁺/ mCherry⁺ and single mCherry⁺ representing active and inactive HIV-1 promoters, respectively, in LTatCL[M] transfected cells. The in vivo observed preferential HIV-1 integration loci in BACH2, BACH2_i5s, is highlighted by a red box.

| LTatCL[M] integration site and orientation | HIV-1 promoter phenotype[1] | Monoclonal cell lines (*n*) | Targeted integration confirmed by PCR (%) | Mono-allelic targeted integration (%) | Unaltered *BACH2* mRNA expression (%) | Unaltered BACH2 protein expression (%) |
|---|---|---|---|---|---|---|
| *BACH2_i5s* | Active | 6 | 5/6 (83.3) | 5/5 (100) | 2/2 (100) | 5/5 (100) |
| | Inactive | 7 | 5/7 (71.4) | 5/5 (100) | n.a. | 4/4 (100) |
| *BACH2_i5c* | Active | 4 | 4/4 (100) | 4/4 (100) | 2/2 (100) | 4/4 (100) |
| | Inactive | 1 | 1/1 (100) | 1/1 (100) | n.a. | 1/1 (100) |
| *BACH2_i2s* | Active | 7 | 7/7 (100) | 7/7 (100) | 2/2 (100) | 7/7 (100) |
| | Inactive | 4 | 3/4 (75) | 3/3 (100) | n.a. | 3/3 (100) |
| *BACH2_i2c* | Active | 4 | 4/4 (100) | 4/4 (100) | 2/2 (100) | 4/4 (100) |
| | Inactive | 2 | 2/2 (100) | 2/2 (100) | n.a. | 2/2 (100) |
| AAVS1_s | Active | 2 | 2/2 (100) | 2/2 (100) | 1/1 (100) | 2/2 (100) |
| | Inactive | 3 | 3/3 (100) | 3/3 (100) | n.a. | n.a. |
| AAVS1_c | Active | 3 | 3/3 (100) | 3/3 (100) | 2/2 (100) | 3/3 (100) |
| | Inactive | n.a. | n.a. | n.a. | n.a. | n.a. |
| Total | | 43 | 39/43 (90.7) | 39/39 (100) | 11/11 (100) | 35/35 (100) |

**Notes:**
[1] Active= Cerulean⁺/mCherry⁺, inactive = single mCherry⁺
n.a. = not available

First, targeted integration of the HIV-1-based, dual-fluorophore vector LTatCL[M] was verified by amplification of a fragment spanning the targeted gene up- or downstream of the 5′ or 3′ homologous arms, respectively, and LTatCL[M]. Targeted integration was confirmed in 39/43 monoclonal cell lines representing all three genomic loci in both orientations and for both fluorescence phenotypes, except for AAVS1_c single mCherry⁺ (Table 1).

Second, targeted integration of LTatCL[M] in those 39 monoclonal cell lines was verified to be mono-allelic by a PCR strategy that allows sufficient amplification only of the not targeted *BACH2* or AAVS1 allele by using primers up- and downstream of the *BACH2* or AAVS1 homologous arms (Fig. 1A) and short PCR extension times. All monoclonal cell lines contained a not targeted allele of the respective targeted integration site (Table 1).

Third, *BACH2* mRNA expression was quantified at 120 days post sorting in 11 Cerulean⁺/mCherry⁺ monolonal cell lines, representing each targeted integration variant. A decrease of *BACH2* mRNA expression could not be observed in any Cerulean⁺/ mCherry⁺ monolonal cell line (Fig. 2A; Table 1).

Forth, the potential impact on BACH2 protein expression after targeted integration of LTatCL[M] into *BACH2* was investigated by Western blot analyses at 120 days post sorting. In line with unaltered *BACH2* mRNA expression, BACH2 protein expression was not impaired compared to non-transfected Jurkat T-cells in all 35 monoclonal cell lines tested (Fig. 2B; Table 1).

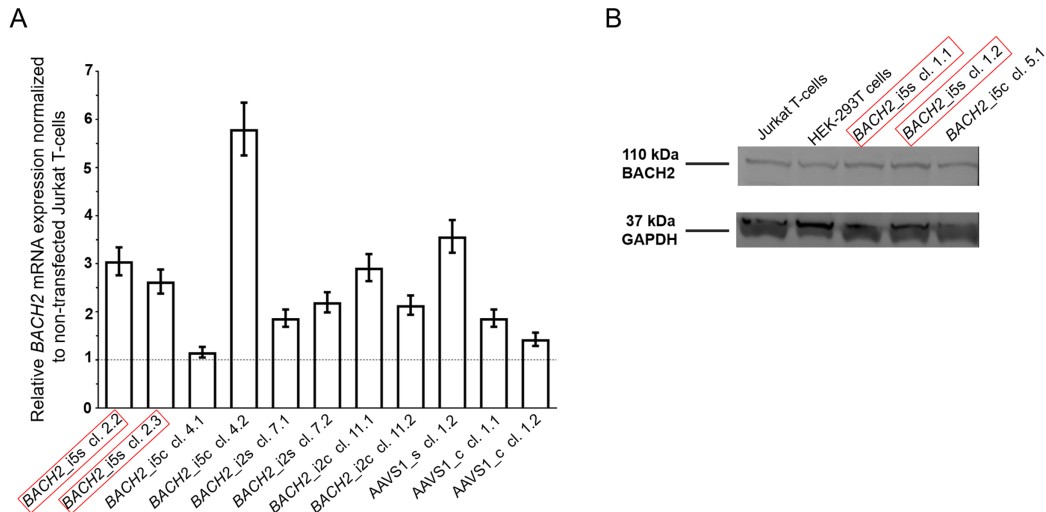

**Figure 2 The mono-allelic integration of LTatCL[M] into *BACH2* did not impair *BACH2* mRNA nor BACH2 protein expression measured 120 days post sorting.** (A) *BACH2* mRNA expression was quantified by amplification of a region in the mRNA downstream of *BACH2*_i5 and *BACH2*_i2. The bar blot consists of two independent qPCR experiments, consisting of two replicates each. The qPCR was carried out two times independently each in duplicates. The means and standard deviations are depicted. One-way ANOVA was applied on all cell clone variants and showed no significance ($p = 0.80$). (B) Western blot analysis of BACH2 protein expression, 110 kD BACH2 (upper panel) and 37 kD GAPDH (lower panel) are shown for one exemplary experiment. The in vivo observed preferential HIV-1 integration loci in *BACH2*, BACH2_i5s, is highlighted by red boxes.

Fifth, HIV-1 integration can affect various aspects of cellular physiology, for instance, cell proliferation. To test whether targeted integration of LTatCL[M] into *BACH2* would lead to an evolutionary advantage in cell growth, a cell-growth competition assay was performed. The stably Cerulean and mCherry expressing *BACH2*_i5s_1.1 Cerulean$^+$/mCherry$^+$ and AAVS1_s_2.1 Cerulean$^+$/mCherry$^+$ monoclonal cell lines were combined in an approximately 1:1 ratio with the parental Jurkat T-cell line and cocultured for 25 days. A comparable decrease of Cerulean$^+$/mCherry$^+$ expression was observed for both the *BACH2*_i5s_1.1 Cerulean$^+$/mCherry$^+$ and the AAVS1_s_2.1 Cerulean$^+$/mCherry$^+$ monoclonal cell line (Fig. S5).

In summary, we successfully performed targeted integration of the 5.3 kb HIV-1-based, dual-fluorophore vector LTatCL[M] into various loci of the human genome via CRISPR/Cas9-mediated homology-directed repair. We subsequently generated monoclonal cell lines modelling actively and latently HIV-1-infected cells with integrated LTatCL[M] in either transcriptional orientation in the targeted genomic loci.

## Silencing of the HIV-1 promoter in Jurkat T-cells with integrated LTatCL[M] in BACH2

To study the HIV-1 promoter activity over time in Cerulean$^+$/mCherry$^+$ monoclonal cell lines, that is active HIV-1 promoter, the fluorescence profile was frequently monitored for 162 days. In the majority of Cerulean$^+$/mCherry$^+$ monoclonal cell lines with integrated

A

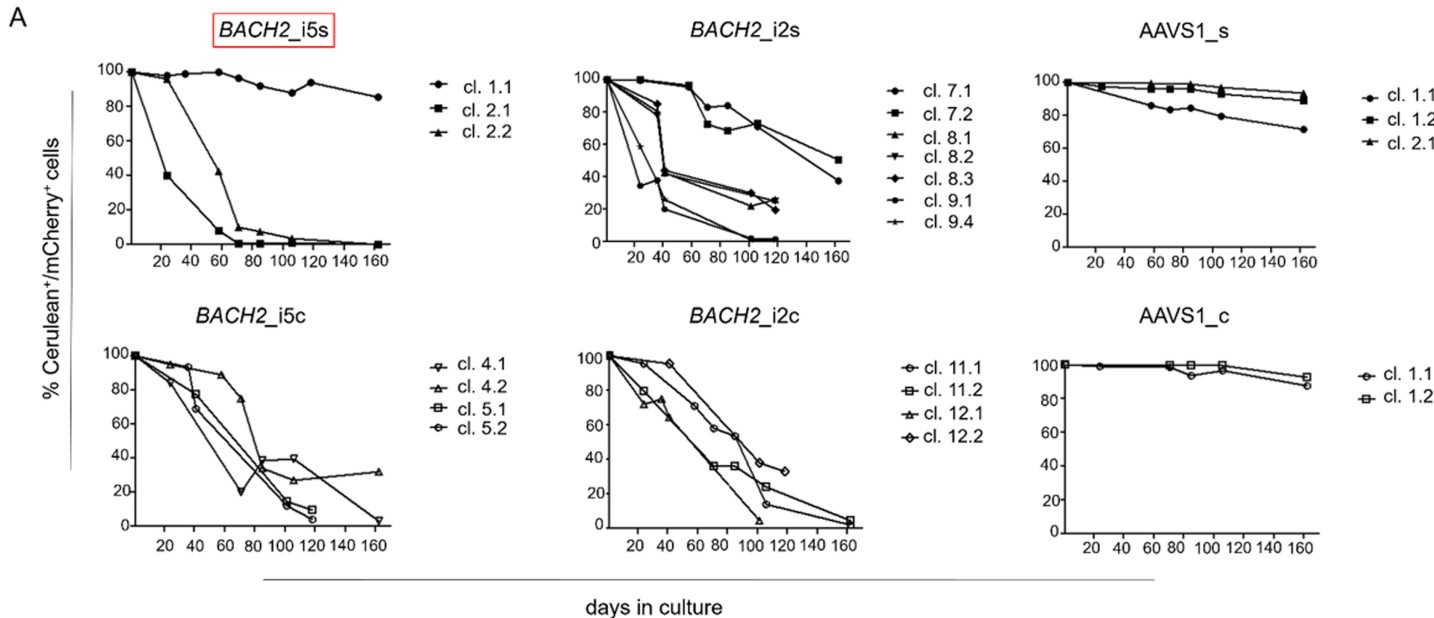

B

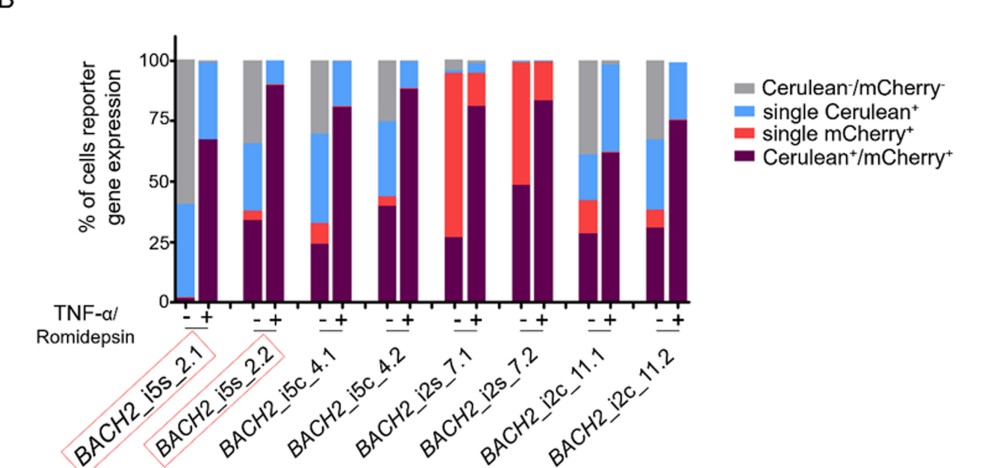

**Figure 3 Longitudinal analysis of Cerulean⁺/mCherry⁺ monoclonal cell lines and reactivation of silenced HIV-1 promoter.** (A) Longitudinal frequency of Cerulean⁺/mCherry⁺ expression in 24 monoclonal cell lines for up to 162 days: 3x *BACH2*_i5s (depicted in red, first left panel), 4x *BACH2*_i5c, 7x *BACH2*_i2s, 4x *BACH2*_i2c, 3x AAVS1_s, and 2x AAVS1_c. Each symbol represents one Cerulean⁺/mCherry⁺ monoclonal cell line. (B) Reactivation of silenced HIV-1 promoter upon treatment of monoclonal cell lines with 10 ng/µL TNF-α and 4 nM Romidepsin for 24 hrs followed by FACS analysis. The experiment was carried out three times independently. The in vivo observed HIV-1 integration loci in *BACH2*, *BACH2*_i5s, is highlighted by red boxes.

LTatCL[M] in *BACH2*, the HIV-1 promoter was gradually silenced as observed by the decline of the frequency of Cerulean⁺/mCherry⁺ cells to <50% in 16/18 monoclonal cell lines within 24–162 days and to <10% in 10/18 monoclonal cell lines within 58–162 days (Fig. 3A; Fig. S1). This was observed independent of the *BACH2* introns 2 or 5 chosen for targeted integration of LTatCL[M] and the transcriptional orientation of LTatCL[M] relative to *BACH2*. In contrast, in all 5 AAVS1_s and AAVS1_c Cerulean⁺/ mCherry⁺ monoclonal cell lines the frequency of Cerulean⁺/mCherry⁺ cells remained

relatively stable for 162 days (>80% in 4/5 cell clones) as observed for one *BACH2*_i5s clone (Fig. 3A). Overall, the HIV-1 promoter when integrated into the *BACH2* gene is silenced over time in the majority of Cerulean⁺/mCherry⁺ monoclonal cell lines whereas it remains active when integrated into AAVS1.

### Silenced HIV-1 promoters can be reactivated by TNF-α and Romidepsin

To evaluate the reactivation of silenced HIV-1 promoters in Cerulean⁺/mCherry⁺ monoclonal cell lines, which gradually lost Cerulean expression, we performed an activation assay using TNF-α and Romidepsin. TNF-α and Romidepsin in combination have been shown to activate the silenced HIV-1 promoter (*Kok et al., 2018*; *Sogaard et al., 2015*). These compounds were used to evaluate the reactivability of the silenced HIV-1 promoter in 8 Cerulean⁺/mCherry⁺ monoclonal cell lines with integrated LTatCL[M] in BACH2. After 99 days, 1.6–48.2% of cells were Cerulean⁺/mCherry⁺. Upon treatment with TNF-α and Romidepsin, the frequencies of Cerulean⁺/mCherry⁺ cells increased to 61.6–89.8% (Fig. 3B), demonstrating that the silenced HIV-1 promoter was reactivatable in those monoclonal cell lines.

### Monoclonal single mCherry⁺ cell lines harbour large internal deletions in LTatCL[M]

A total of 17 single mCherry⁺ monoclonal cell lines, presumably representing latently HIV-1-infected cells, silenced at an early time point upon targeted HIV-1 integration, were treated with TNF-α and Romidepsin to activate the HIV-1 promoter. Cerulean expression was not induced by TNF-α and Romidepsin in any of these 17 monoclonal cell lines (Fig. S1). To further investigate this, the whole vector LTatCL[M] was amplified and sequenced in 8 of these 17 monoclonal cell lines. Surprisingly, the integrated vector LTatCL[M] in these monoclonal single mCherry⁺ cell lines harboured large internal deletions in the 5'HIV-1 LTR, tat and/or Cerulean cassette (Fig. 4). The monoclonal *BACH2*_i5s cell lines 3.1 and 3.2, which were derived from the same cell culture after the first sort, showed the same deletion in the IRES-Cerulean region (Fig. 4A). A similar observation was made in the monoclonal *BACH2*_i2c cell lines 13.1 and 13.3, also derived from the same first sorted cell population: The same deletion spanning from the 5′LTR to the Cerulean cassette was observed in both monoclonal cell lines (Fig. 4B). All independent single mCherry⁺ monoclonal cell lines contained large deletions in different regions of the vector suggesting that the plasmids were not the source of those deletions. This was confirmed by next-generation sequencing of the plasmids not showing any evidence for large deletions. Next, the integrated vector LTatCL[M] was sequenced in a cell population of single mCherry⁺ cells immediately after the first sort, 9 days post transfection (Fig. S1). Evidence for numerous different large deletions were observed in the 5'HIV-1 LTR, Tat, IRES and/or Cerulean (Fig. S1), indicating that those deletions occurred early during targeted integration.

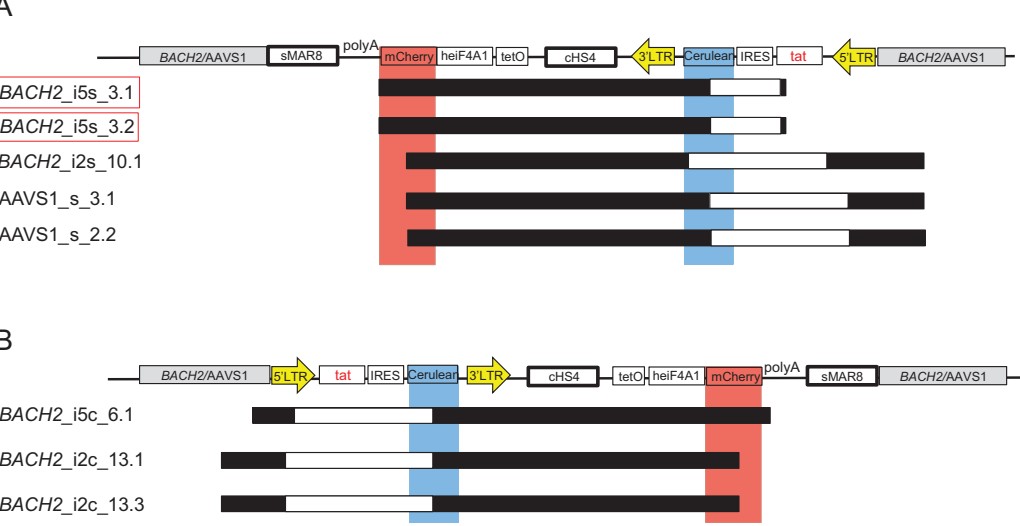

**Figure 4 Mapping of large deletions within LTatCL[M] in TNF-α/Romidepsin non-responsive, single mCherry⁺ monoclonal cell lines.** Deletions in the integrated vector LTatCL[M] are depicted for monoclonal *BACH2*_i5, *BACH2*_i2 and AAVS1 monoclonal cell lines with integrated LTatCL[M] in the same transcriptional orientation (A) and in the convergent transcriptional orientation (B). Black horizontal bars show the amplified and sequenced regions of LTatCL[M] and large deletions are shown in white horizontal bars. Light blue and light red vertical bars show the positions of the two fluorophores Cerulean and mCherry, respectively. Vector schemes are depicted on top. The in vivo observed HIV-1 integration region site in *BACH2*, *BACH2*_i5s, is highlighted by red boxes. Different starting points of amplicons were due to using different primer pairs for amplifications.

# DISCUSSION

We developed a novel model to study HIV-1 promoter activity based on CRISPR/Cas9-mediated targeted integration of an 5'337 bp HIV-1-based, dual-fluorophore vector into selected sites in the human genome. In our previous study, we have shown that the HIV-1-based, dual-fluorophore vector LTatC[M] reproduces features of active and latent HIV-1 infections (*Kok et al., 2018*). Here, we studied the fate of the HIV-1 promoter in specific HIV-1 integration loci/sites by targeted integration of the vector LTatCL[M] into *BACH2* and AAVS1. In Jurkat T-cells, HIV-1-based vector integration into *BACH2* led initially to an active HIV-1 promoter as shown by the expression of HIV-1 LTR controlled Cerulean. Over time those monoclonal cell lines showed a gradual silencing of the HIV-1 promoter. This might be due to transcriptional interference, which can occur in two ways: Either through promotor occlusion or convergent transcription, in which the transcription from the host genes interferes with HIV-1 transcription (*Han et al., 2008*; *Lenasi, Contreras & Peterlin, 2008*; *Shan et al., 2011*). In our study, however, HIV-1 integration into the two target loci of *BACH2* intron 5 and intron 2 in both orientations showed similar silencing of the HIV-1 promoter over time. Differences were observed for the genomic safe harbour AAVS1, in which no substantial silencing of the HIV-1 promoter was observed over a time period of 162 days. This raises the question, whether *BACH2* is an exceptional HIV-1 integration site promoting HIV-1 promoter silencing or, vice versa, AAVS1 is an exceptional HIV-1 integration site preventing HIV-1

promoter silencing. In a future study, we will be expanding the repertoire of investigated HIV-1 integration sites.

In virally suppressed HIV-1-infected individuals, the provirus integrated into *BACH2* has been found predominantly in intron 5 in the same transcriptional orientation as the gene and has been linked with clonal expansion. However, in in vitro in HIV-1 infected cell lines, integration has been found to occur randomly in the *BACH2* gene, indicating that the integration selection observed in *vivo* cannot be fully recapitulated (*Maldarelli et al., 2014*; *Wagner et al., 2014*). The preference of HIV-1 integration into *BACH2* intron 5 in the same orientation in vivo could be caused by: (1) Selection of *BACH2* intron 5 integrants over other integrants over time (*Hughes & Coffin, 2016*), (2) *BACH2* intron 5 is preferentially targeted by HIV-1 in primary CD4$^+$ T-cells, presumably due to the spatial location of this locus in the nucleus of target cells (*Marini et al., 2015*), or (3) distinct alterations in *BACH2* expression levels caused by different HIV-1 integration patterns. *BACH2* expression levels vary between T-cells in distinct differentiation stages (*Richer, Lang & Butler, 2016*). An increase of BACH2 transcripts in regulatory and effector T-cells were observed when HIV-1 was integrated in *BACH2* (*Cesana et al., 2017*). Enhanced expression of the wild-type *BACH2* in regulatory T-cells leads to increased proliferation capacity without affecting the cells' phenotype (*Cesana et al., 2017*). Alteration of *BACH2* expression might cause HIV-1 persistence in *BACH2* intron 5 and expansion of the cell. In Jurkat T-cells, as compared to primary CD4$^+$ T-cells, the *BACH2* expression levels might be different (*Shan et al., 2017*). We measured *BACH2* expression 120 days post sorting. At this time point the majority of the transfected cell clones harboured silenced HIV-1 promoters. We could not observe an inhibitory effect on *BACH2* mRNA or protein expression when *BACH2* introns 5 and 2 were targeted for integration with our HIV-1-based vector. This indicates that other factors might lead to the persistence of HIV-1 in this locus in Jurkat T-cells. HIV-1 infection into a cellular gene can affect various aspects of cellular physiology as for example proliferation or a longer half-life of the cell (*Rezaei & Cameron, 2015*; *Wagner et al., 2014*). However, there was no evidence that targeted integration of LTatCL[M] into *BACH2* had an evolutionary advantage in cell growth in competition with the parental Jurkat T-cell line.

The expression of Cerulean decreased over time in almost all *BACH2* monoclonal cell clones independent of the targeted intron and the orientation of HIV-1-based vector integration. The slow decline and the presence of Cerulean positive cells for up to 162 days in some of those monoclonal cell clones shows that targeted integration of LTatCL[M] into *BACH2* does not immediately and not completely lead to HIV-1 promoter silencing in Jurkat T-cells. However, insulator-protected mCherry expression also declined in some of those cell clones, suggesting a gradual and strong silencing effect at the *BACH2* loci. This silencing was reversible as shown by the treatment of cell clones harbouring silenced HIV-1 promoters with TNF-α and Romidepsin that led to HIV-1 promoter reactivation. These observations are partly in line with *Lange et al. (2018)*: They observed a comparable reversible silencing of the HIV-1 promoter in *BACH2* within 20–40 days targeting two sites within *BACH2* intron 5 via CRISPR/Cas9 and inserting one reporter gene under the control of the HIV-1 LTR-promoter in the same transcriptional orientation

as *BACH2*. Here, we observed the same phenotypes in another intron of *BACH2* and independent of the orientation of the HIV-1 integration. Furthermore, targeted integration of LTatCL[M] into the genomic safe harbour AAVS1 did not lead to silencing of the HIV-1 promoter. This model can be further expanded by insertion of, for instance, certain HIV-1 genes, which will allow to study the impact of viral genes on the host's expression profile and cell cycle.

Mono-allelic integration of 5.3 kb long LTatCL[M] into *BACH2* and AAVS1 has been confirmed in monoclonal cell lines by amplifying and sequencing junctions of integration. So far, the longest fragment inserted in human lymphocytes was 1.5 kb long (*Hung et al., 2018*), hence, our findings show that insertion of a 3.5x larger fragment is possible in Jurkat T-cells. Fragments of similar length (5.5 kb and 7.4 kb) have been successfully integrated in other cell types (embryonic stem cells and zygotes) (*Wang et al., 2015*).

Monoclonal cell lines expressing initially only mCherry, that is supposed to model latently HIV-1 infected cells, were found to contain large deletions in the LTR-tat-Cerulean cassette. Similar deletions were already observed by us when we used a variant of our vector to generate retroviruses for infection of cells (*Kok et al., 2018*). There, we speculated that the deletions were caused by error-prone HIV-1 reverse transcription (*Bebenek et al., 1989*; *Patel & Preston, 1994*) or copy-choice recombination (*Sanchez et al., 1997*; *Simon-Loriere & Holmes, 2011*), which is also observed in cells from HIV-1-infected individuals (*Ho et al., 2013*). However, our current system does not require reverse transcriptase prior to integration since we are using CRISPR/Cas9-mediated targeted integration. Therefore, the deletions might be due to recombination events triggered by the LTRs flanking the Cerulean cassette (*Mager & Goodchild, 1989*). Together with our previous observation (*Kok et al., 2018*), our results underline the importance to further investigate single fluorescent cell populations (representing latently HIV-1 infected cells) in latency models using HIV-1-based vector systems. Beyond HIV-1, these findings might also have implications for lentiviral vectors used for gene therapy.

## CONCLUSION

Using the CRISPR/Cas9-technology, stable targeted integration of the 5.3 kb long HIV-1 based, dual-fluorphore vector LTatCL[M] was successful, enabling longitudinal studies on effects of selected genomic sites on HIV-1 promoter activity and cellular phenotypes. Targeting the *BACH2* gene revealed that loci within this gene are capable of supporting an active HIV-1 promoter upon integration but its activity diminishes over time in Jurkat T-cells. On the contrary, the HIV-1 promoter was not silenced when integrated into the genomic safe harbour AAVSI of Jurkat T-cells.

## LIST OF ABBREVIATIONS

**ART**      Antiretroviral therapy
**BACH2**    BTB domain and CNC homology 2
**AAVS1**    Adeno-associated virus integration site 1
**LTR**      Long terminal repeat

**LTatCL[M]** 5′LTR—Tat-Cerulean-3′LTR-insulator-mCherry-insulator
**TNF-α** Tumor necrosis factor alpha
**RIG** Recurrent integration gene

## ACKNOWLEDGEMENTS

We are very thankful to Walter Schaffner (Institute for Molecular Life Science, UZH) for fruitful discussions and critical reading of the manuscript. We thank Philipp Schätzle and Andrea Henning (Flow Cytometry Facility, UZH, Zurich, Switzerland) for bulk and single cell sorting services and Stefan Schmutz for technical assistance. The following reagent was obtained through the NIH AIDS Reagent Programme, Division of AIDS, NIAID, NIH: Sup-T1 from Dr. Dharam Ablashi. This manuscript has been released as a pre-print at bioRxiv (*Inderbitzin et al., 2020*).

### Funding

This study was funded by the Swiss National Science Foundation, grant No. 310030_141067/1 to Karin J Metzner and from the Forschungskredit Candoc, grant No. FK-19-032 to Anne Inderbitzin. The funders had no role in study design, data collection and analysis, decision to publish, or preparation of the manuscript.

### Grant Disclosures

The following grant information was disclosed by the authors:
Swiss National Science Foundation: 310030_141067/1.
Forschungskredit Candoc: FK-19-032.

### Competing Interests

Karin J. Metzner has received travel grants and honoraria from Gilead Sciences, Roche Diagnostics, Tibotec, Bristol-Myers Squibb, and Abbott; the University of Zurich has received research grants from Gilead, Roche and Merck Sharp & Dohme for studies that Karin J. Metzner serves as principal investigator, and advisory board honoraria from Gilead Sciences. The authors declare that the research was conducted in the absence of any commercial or financial relationships that could be construed as a potential conflict of interest. All other authors declare no competing interests relevant to this study.

### Author Contributions

- Anne Inderbitzin conceived and designed the experiments, performed the experiments, analysed the data, prepared figures and/or tables, authored or reviewed drafts of the paper, and approved the final draft.
- Yik Lim Kok conceived and designed the experiments, performed the experiments, analysed the data, prepared figures and/or tables, authored or reviewed drafts of the paper, and approved the final draft.
- Lisa Jörimann performed the experiments, analysed the data, prepared figures and/or tables, authored or reviewed drafts of the paper, and approved the final draft.
- Audrey Kelley performed the experiments, authored or reviewed drafts of the paper, and approved the final draft.
- Kathrin Neumann performed the experiments, authored or reviewed drafts of the paper, and approved the final draft.
- Daniel Heinzer performed the experiments, authored or reviewed drafts of the paper, and approved the final draft.
- Toni Cathomen conceived and designed the experiments, authored or reviewed drafts of the paper, and approved the final draft.
- Karin J. Metzner conceived and designed the experiments, analysed the data, authored or reviewed drafts of the paper, and approved the final draft.

## Data Availability

All data is available on Zenodo:

Anne Inderbitzin, Yik Lim Kok, Lisa Meret Joerimann, Audrey Kelley, Kathrin Neumann, Daniel Heinzer, Toni Cathomen, Karin J. Metzner. (5 August 2020). HIV-1 promoter is gradually silenced when integrated into BACH2 in Jurkat T-cells. Zenodo. DOI 10.5281/zenodo.3972946.

## Supplemental Information

Supplemental information for this article can be found online at http://dx.doi.org/10.7717/peerj.10321#supplemental-information.

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
