# Peer review of "HIV-1 promoter is gradually silenced when integrated into BACH2 in Jurkat T-cells"

_PeerJ, doi:10.7717/peerj.10321_

## Round 0.1 · original submission · Minor Revisions

As you will see, both reviewers found your study interesting and well-performed. Both have indicated some changes are required. Reviewer-1 has some comments that do not require experimental work, but reflect some caveats about the cell line used in the study. Please address these in the Discussion so that these are clearly sign-posted to the readership.

Reviewer-2 has a slightly longer list of points and some experimental work is suggested to address a few issues of interpretation; these would be good, if the data is either available or can be readily produced. However, this reviewer is of the mind that suitable discussion of these potential limitations should represent an acceptable alternative. I leave this to you.

Please provide a clear and detailed rebuttal/explanation of changes, so that I can expedite this rather than send it back out to review. I enjoyed reading your paper.

·

Basic reporting

In their manuscript entitled “HIV-1 promoter is gradually silenced when integrated into BACH2 in Jurkat T-cells, Inderbitzin and colleagues report the development of initial characterization of a system to investigate the impact of integration site choice and HIV gene expression (latency). I found the introduction to be clearly written and to explain the knowledge gap being addressed. The Figures were easy to follow and the Materials and Methods section is detailed enough, in my opinion, to allow independent replication of the experiments. Overall the English is high quality and accessible for international readers.

Experimental design

The authors employ dual reporter viruses that have previously been published from their group. These constructs were integrated into the genome of Jurkat T cells using CRISPR/Cas9 technology via the addition of arms for homologous recombination into the known BACH2 gene that has been shown by several independent groups to be a “hot spot” for HIV integration.

The work certainly suits the broad scope of the journal and will be of use to others working in the field. One question for the authors, in the section “Availability of materials” the authors have marked not applicable. Given the use of new viral constructs and the creation of useful cell lines in the work, why is this the case?

Validity of the findings

The experiments are in my view well controlled and the results are compelling. The authors show that HIV expression is gradually reduced in the BACH2 locus over time in an orientation-independent fashion, without significantly disturbing the expression of BACH2 protein levels. These results contrast with the integration into AAVS1 “safe haven” genomic site where HIV remains active over time. It should be fascinating to use the system in the future to further explore other genomic sites.


Although I found the authors discussion of the results to be reasonable, I had two minor points that should be addressed for clarity to readers. Potential limitations of the system include the fact that an immortalized cancer cell line was employed. Given the fact that genomic preferences are only found in primary cells, certain elements of the system may not recapitulate the selection seen in vivo. A second limitation that should be mentioned is the fact that other viral genes such as vpr are not in these constructs. Since cell cycle impacts HIV gene expression and latency, and vpr manipulates the cell cycle, the absence of these genes could have importance on the future findings.

Additional comments

Overall, I commend the authors on a significant amount of careful work and a useful contribution to the field.

Reviewer 2 ·

Basic reporting

Inderbitzin et al. presents a study of HIV-1 integration into introns of the BACH2 gene in Jurkat cells. The author’s approach is using CRISPR insertion of a dual HIV-1 reporter, one gene reporting HIV-1 LTR activity, and another constitutively active. Authors insert this vector into two introns of BACH2 in sense and antisense orientation (relative to the gene) as well as an alternate locus in Jurkat cells. Authors then iteratively sort integrated cells to obtain monogenic clones. Authors observe a limited impact on the gene, BACH2, in terms of RNA and protein expression. Authors also observe a gradual decline in the expression of the HIV specific reporter. While a very interesting study, the approach is limiting to the general applicability of the results and interpretation must be made with caution. The authors should address that it is possible that the finding is because of insulator insertion rather than HIV insertion into BACH2 (if not experimentally justified).

Experimental design

Overall the authors spent respectful effort in inserting this cassette into the BACH2 site with several controls.

Validity of the findings

1. Investigators have used a reporter with genomic insulating elements flanking promoters. While desirable for generating a dual reporter, authors have not demonstrated that these do not impact the ability of the HIV-1 LTR from impacting the integrated gene. The cHS4 element is specifically investigated in viral vector gene therapy because of its ability to block chromatin modifications and enhancer promoter interactions (see Emery et al. PNAS 2000). Enhancer blocking activity was later discovered to be due to a CTCF binding site (Bell et al. Cell 1999), which has been implicated gene expression changes in other human retroviruses (Satou et al. PNAS 2016, Melamed et al. eLife 2018). Given these known impacts, it is unclear that authors have demonstrated the impact of the HIV-1 LTR on the inserted gene. To make this claim, authors would need to generate clones without these insulator elements. Further, the vector used in this study contains elements that would likely interfere with read through HIV-1 driven transcription, such as the additional reporter promoter and the polyadenylation site. Or, alternatively, can the authors address these limitations of having significant transcriptional interference by the insulator cHS4, that the results of may be caused by insulator insertion?
2. Authors did not functionally validate the impact of HIV-1 integration into BACH2. While changes may not be appreciable at the qPCR and western blot level, integration into BACH2 may provide additional competitive advantages or clonal expansion. Authors should conduct competition experiments to determine if this is the case. Authors could perform these experiments either directly (by mixing parental cells with integrant cells and watching the ratio of parental to integrant cells overtime) or indirectly (by starting several wells of clones with the same number of cells and plotting expansion over time. Or, alternatively, can the authors address these caveats in their results?
3. Authors could strengthen their claim that this system is physiologically relevant by showing their system recapitulates production of known chimeric RNAs such as those reported in Cesana et al. Nature Communications 2017 and Liu et a., Science Translational Medicien 2020. Or, alternatively, can the authors address this caveat, that without knowing whether this HIV inserts is still able to produce aberrant splicing to BACH2, it is unclear that whether the phenotype observed is because of HIV integration into BACH2 or the insulator insertion into BACH2?
4. Investigators are unclear when they state “BACH2 mRNA nor BACH2 protein expression measured 120 days post sorting”. In figure 1D authors show these experiments occurring 25-30 days post second sort, which is 50 days post initial transfection which only adds to 75-80 days. One of the numbers included must be incorrect, and should be rectified for consistency and clarity. Can the authors revise?

---

## Round 0.2 · accepted · Accept

Thank you for your considered responses; I am happy to now accept the paper for publication.